# Medical Face Masks Do Not Affect Acid–Base Balance Yet Might Facilitate the Transmission of *Staphylococcus aureus* in Hospital Settings during the COVID-19 Pandemic

**DOI:** 10.3390/ijerph20032474

**Published:** 2023-01-30

**Authors:** Piotr Ostrowski, Helena Masiuk, Piotr Kulig, Anastasiia Skoryk, Aleksandra Wcisłek, Joanna Jursa-Kulesza, Angela Sarna, Michał Sławiński, Maciej Kotowski, Karol Tejchman, Katarzyna Kotfis, Jerzy Sieńko

**Affiliations:** 1Department of General Surgery and Transplantation, Pomeranian Medical University, 70-111 Szczecin, Poland; 2Independent Laboratory of Medical Microbiology, Pomeranian Medical University, 70-111 Szczecin, Poland; 3Department of General Pathology, Pomeranian Medical University, 70-111 Szczecin, Poland; 4Department of Laboratory Diagnostics, Public Clinical Hospital No. 2, 70-111 Szczecin, Poland; 5Department of Anesthesiology, Intensive Therapy and Acute Intoxications, Pomeranian Medical University, 70-111 Szczecin, Poland

**Keywords:** face masks, COVID-19 pandemic, staphylococcus aureus, acid–base balance

## Abstract

Introduction: Due to the SARS-CoV-2 coronavirus pandemic, the wearing of masks has become a common phenomenon. Most of the undesirable effects of using a protective face covering are usually related to the prolonged time of its wearing, and the adverse consequences of face coverings should be considered two-fold. The aim of the study was to evaluate the rate of contamination of the three types of face coverings (surgical, N95, and FFP2 masks) with the microorganism—aerobic bacteria, yeasts, and molds—after the 3 h exposure time. The study aimed to investigate the effects of wearing FFP2 masks (KN95) on respiratory function and the acid–base balance of the human body. Results: The presence of *S. aureus* was confirmed in both nasal carriers and non-carriers which may demonstrate the cross-contamination and spread of this bacterium via hands. *S. aureus* was found on external and internal surfaces of face masks of each type, and therefore could also be transmitted via hands from external sources. The 3 h exposure time is not sufficient for Gram-negative rods and mold contamination. Moreover, there were no significant differences in most of the parameters studied between the first and second examinations, both in spirometry and capillary blood gas analysis (*p* > 0.05).

## 1. Introduction

Severe acute respiratory syndrome coronavirus-2 (SARS-CoV-2) is a novel RNA virus responsible for the global COVID-19 pandemic. The very first case was officially reported in December 2019 in the Chinese city of Wuhan [1]. Since then, the virus rapidly spread across the world, becoming a great challenge for society around the world, particularly the healthcare system. The symptomatology of the disease is complex. The infection can be asymptomatic; can cause flu-like symptoms, which may differ in severity; and can even lead to fatal respiratory distress accompanied by multiorgan failure with underlying hyperinflammatory responses and cytokine storm [2]. Nonetheless, the most observed initial manifestation of COVID-19 resembles any viral infection that affects the upper respiratory tract, including fever, cough, breathing difficulties, sore throat, fatigue, as well as gastrointestinal symptoms [3]. A loss of smell and taste could be considered as a peculiar manifestation of COVID-19, particularly during the initial stage of the infection [4].

To tackle the disease, the worldwide authorities introduced safety measures which were supposed to reduce the spread of the virus. One of the most common regulations was the recommendation of protecting the airways. Many types of protective personal equipment (PPE) were introduced for the face, from cloth masks to specialized FFP3 respirators. While the public predominantly used cloth or surgical masks, healthcare professionals, especially within hospital settings, tended to use more advanced face coverings. During the COVID-19 pandemic, face mask use by the public became an unprecedented phenomena, including potentially vulnerable populations such as children, elderly, and chronically ill individuals. It was suggested that prolonged mask usage can cause several adverse effects and worsen users’ overall condition [5]. 

Adverse face skin reactions following the use of different types of masks were noted among general population during the COVID-19 pandemic [6]. Therefore, the duration of interrupted face cover was also recognized. Furthermore, inappropriate mask wearing was also found to affect the skin and negatively influence the skin microbiota [7]. Moreover, prolonged and uninterrupted airway coverage may alter skin moisture and limit oxygen supply, leading to the breakdown of the epidermal barrier. Although it has been proven that most of the bacteria contaminating masks originate from the skin and respiratory tract microbiota [8], prolonged mask usage could worsen pre-existing dermatoses, acne flare-ups, and skin abrasions by becoming a gateway for bacterial and fungal contamination [9]. 

The available literature on the pandemic period suggests a new term in relation to disorders of facial skin function, resulting from the mechanical irritation of its surface by covering the face—maskne—derived from conjunction of the words mask and acne [10]. Face skin dysbiosis may result in exacerbations of skin diseases including eczema, perioral dermatitis, and rosacea [11]. Moreover, increases in the temperature and humidity of the skin covered with a mask were also considered to significantly affect its functioning. Spigariolo et al. suggested that such a microclimate may substantially influence the skin microbiota composition [10].

The microorganisms present in a hospital environment can pose a potential threat to public health, and their cross-transmission through PPE may contribute to an increase in the incidence of nosocomial infections. In 2017, the WHO published a list of ESKAPE pathogens that posed the greatest challenges to modern medicine [12]. The abbreviation ESKAPE includes six nosocomial microorganisms that demonstrate multidrug resistance and represents therapeutic challenges especially during the COVID-19 pandemic. The ESKAPE group includes Enterococcus faecium, Staphylococcus aureus, Klebsiella pneumoniae, Acinetobacter baumannii, Pseudomonas aeruginosa, and Enterobacter spp. The persistent use of antibiotics has resulted in the emergence of multidrug-resistant (MDR) and extensively drug-resistant (XDR) bacteria, which make even the most potent antimicrobial agents ineffective. Carbapenemase-producing Enterobacterales (CPEs) have now become an essential therapeutic issue; thus, novel therapeutic management strategies are needed to treat infections caused by these pathogens. Several alternative therapy approaches have been widely discussed, such as the use of drugs in combination with some novel beta-lactamase inhibitors [8]. 

The duration of wearing the face covering directly correlates with its contamination; thus, the risk of self and the environmental transmission of bacterial and fungal shedding increase with time of face covering exposure. Therefore, questions could be raised regarding the masks’ safety with special emphasis on the fact that face coverings may become a potential source of pathogens in hospital settings when inappropriately worn, stored, or disposed [13,14,15].

Based upon the abovementioned presumptions, we defined the following aims of the study:-To evaluate the contamination rate of three types of face coverings—surgical, FFP2 (KN95 respirator), and FFP3 masks—used over a period of 3 h;-To evaluate the participants’ *S. aureus* carrier state and its association with masks’ microbiota;-To evaluate the impact of uninterrupted face coverings (surgical, FFP2, and FP3 masks) on the acid–base balance;-To evaluate the impact of uninterrupted face coverings (surgical, FFP2, and FP3 masks) on respiratory tract function with spirometry.

## 2. Materials and Methods

### 2.1. Participants

The participants (*n* = 50) were employees of the hospital in which the study was conducted. All participants were asked to wear each face coverings—respiratory cover equipment (N95 and FFP3) and surgical masks (SMs)—for three consecutive hours (exposure time *t* = 3 h) and to carry out their daily activities. 

### 2.2. Carrier State

Initially, the nasal and throat swabs were collected from all participants (*n* = 50) to evaluate the *S. aureus* carrier state. Nasal swabs and throat swabs were cultured onto mannitol salt agar (MSA2) (bioMerieux, Craponne, France), and onto Columbia agar + 5% sheep blood and Sabouraud gentamicin chloramphenicol agar (bioMerieux), respectively. The nasal carriage was assessed based on the positive reaction of coagulase-positive staphylococci on the MSA2 medium, producing yellow colonies and a surrounding yellow medium (Figure 1).

Species confirmation was performed using the Staphaurex™ latex agglutination test (Thermo Fisher, Waltham, MA, USA) (Figure 2) and based on the ability to coagulate rabbit plasma (Biomed, Lublin, Poland). Additionally, the optochin test was performed to determine the presence of Streptococcus pneumoniae in nasal and throat swabs.

### 2.3. Spirometry and Blood Gas Analysis

In order to evaluate the impact of uninterrupted face coverings on the acid–base balance in every participant, fingertip capillary blood gas analysis was performed, with measurements taken both before the mask was put on and after three consecutive hours before the face covering was taken off. The procedure was repeated three times and three different mask types were examined.

The saturation, pulse, and blood pressure were measured in the same manner as the capillary blood gas analysis, i.e., before the mask was put on and before the face covering was taken off. Measurements were repeated three times and three different mask types were examined.

### 2.4. Culturing

Promptly, after the exposure time, the whole external and internal surface of each mask was separately swabbed with cotton sterile swab inoculated with 0.85% sterile saline.

The external and internal surfaces of SMs were successively imprinted onto Columbia agar + 5% sheep blood, and each surface was provided with sterile Count Tact agar (bioMerieux, Craponne, France) to quantitatively evaluate the level of contamination.

Swabs from external and internal surfaces of all masks were cultured onto Columbia agar + 5% sheep blood (COS), mannitol salt agar (MSA2), and Sabouraud gentamicin chloramphenicol agar (SGC2) (bioMerieux, Craponne, France). The COS swabs were incubated for 24 h in a candle jar at 37 °C and each SGC 2 medium was incubated for 48 h at 37 °C in aerobic conditions, before then being subsequently transferred to RT and incubated for the next 3 days. 

### 2.5. Culture Interpretation

The total bacterial load for both surfaces of each face covering was evaluated as follows: > 100 colony-forming units (CFUs)—incalculable, 50–100 CFUs—numerous, between 50–10 CFUs—quite numerous, and less than 10 CFUs—single (with particular CFU numbers if calculable). 

The bacterial cultures on COS were interpreted after the incubation time (24 h). *S. aureus* species identification was performed using the Staphaurex™ latex agglutination test and on the basis of the ability to coagulate rabbit plasma (Biomed, Lublin, Poland). The presumptive identification of microorganisms was performed based on the colony morphology, the type of hemolysis, and Gram staining. The catalase tests with hydrogen peroxide (2%) and 3% KOH were used to differentiate Gram-positive cocci and Gram-negative bacteria, respectively. The presence of coagulase-negative staphylococci (CNS) was determined based on the negative reaction on the MSA2 medium (a lack of yellow colonies and a surrounding yellow medium) and based on the coagulase-negative result.

The determination of fungi presence was performed based on the growth on SGC2 after 24 and 48 h of incubation in 37 °C in aerobic conditions and additionally after 5 consecutive days in RT under aerobic conditions for yeasts and molds, respectively (5 days in total). 

The presumptive identification of aerobic Gram-positive spore-forming rods (Figure 3) was performed based on Gram staining.

### 2.6. Statistical Analysis

The Wilcoxon signed-rank test was implemented for the continuous variables to assess the differences between the first and second measurements. The CFU was ranked from 0 (not detected) to 4 (incalculable) and analyzed using the Wilcoxon signed-rank test. Fisher’s exact test was implemented to assess the differences between categorical variables. A *p* value < 0.05 was considered statistically significant. All calculations were performed in RStudio. 

## 3. Results

### 3.1. Characteristics of the Study Group

Table 1 depicts the characteristics of the study group. Participants were recruited from the medical stuff of our institution. The vast majority of them were women. The mean age of the participant was 43.88 years. The mean weight and height were 76.59 kg and 1.7 m, respectively. The enrolled individuals tended to be slightly overweight with a mean BMI of 26.49. Furthermore, 30% of the study participants were smokers (15/50) and 46% (23/50) reported having chronic illnesses.

### 3.2. Culturing

The species that were found to contaminate the internal and external surfaces of a given face covering were analyzed qualitatively and quantitatively.

Table 2 depicts an analysis of the species detected on the internal and external surfaces of a given face covering. Bacterial load significantly differed between internal and external surfaces. We did not detect Gram-negative rods, yeasts, and molds on the surface of the analyzed face coverings. 

In every participant, the *S. aureus* and *S. pneumoniae* nasal carriage was tested. The carrier state (Table 3) of Staphylococcus aureus was confirmed in 14% of the participants (*n* = 7). On the other hand, microbiological investigations did not reveal any streptococcus pneumoniae carrier. 

In total, 37 surgical masks, 44 KN95 respirators, and 38 FFP3 respirators were returned by the participants for microbiological evaluation. The presence of coagulase negative staphylococci (CNS) was confirmed both on external and internal surfaces. Moreover, on the internal surfaces of each mask, the presence of oral streptococcus was confirmed. The presence of aerobic Gram-positive spore-forming rods (*Bacillus* spp.) was determined on the external surfaces of the surgical, FFP2, and FFP3 masks. Furthermore, the absence of yeasts and molds was determined after 7 days of incubation in all face covering types. The lack of Gram-negative rods (Enterobacterales and non-fermentative rods) was also confirmed. 

Subsequently, we analyzed the presence of *S. aureus* on the masks’ surface among carriers and non-carriers (Table 4). If the bacteria were detected on any surface of the mask, the mask was considered to be contaminated with *S. aureus*. Firstly, we detected *S. aureus* on the mask surface in both the carrier and non-carrier groups. There was no statistically significant difference in the frequency of *S. aureus* detection on the surface of surgical and FFP3 masks between carriers and non-carriers (*p* > 0.05, Fisher’s exact test). Interestingly, *S. aureus* was more frequently detected on the KN95 mask in the non-carrier group (*p* = 0.04236, Fisher’s exact test). 

### 3.3. Blood Gas Analysis and Basic Parameters

Below are the presented results of the capillary blood gas analysis, in addition to the basic parameters such as saturation, pulse, as well as systolic and diastolic blood pressure.

Results of the spirometry examination (Table 5, surgical mask) did not reveal statistical differences between the first and second measurements. Although there was a significant difference in the hematocrit level, it was not clinically relevant, and values were obtained within the reference range. 

Similarly, in most tested parameters, there were no significant differences (Table 6*—*FFP2 mask). However, in lactate, potassium ion and hematocrit were detected and statistically significant differences were found to be clinically irrelevant in the surgical mask.

The obtained results in the FFP3 mask assessment (Table 7) revealed very similar results to those in the surgical and FFP2 masks. However, in this set of measurements, no statistically significant differences between the first and second examinations were detected. 

Although there were no significant differences between the vast majority of measurements, there is one point to be noted. In all measurements, increased oxygen partial pressure levels were detected, and the hemoglobin concentration was below the reference range. 

### 3.4. Spirometry

Table 8 presents the results of spirometry examination which was performed just before the surgical mask was put on and immediately after it was taken off. There were no significant differences detected between the two examinations.

Table 9 presents the results of spirometry examination which was performed just before the FFP2 mask was put on and immediately after it was taken off. There was a statistically significant difference in PEF between the two examinations.

Table 10 presents the results of the spirometry examination which was performed just before the FFP3 respirator was put on and immediately after it was taken off. There were no significant differences detected between the two examinations.

## 4. Discussion

Spirometry is an examination used for pulmonary function evaluation. It determines lung function based on the amount (volume) and speed of inhaled and exhaled air. It is a clinically useful and reliable tool in the diagnosis of obstructive conditions such as asthma and chronic obstructive pulmonary disease. Nonetheless, it can only suggest restriction which should be further confirmed in body plethysmography [16]. Capillary blood gas (CBG) sampling is a less invasive method and generates less complications than arterial blood gas (ABG) sampling. CBG is also used for the analysis of estimating acid–base balance, oxygenation, and ventilation adequacy. The CBG sample is collected by puncturing the skin in a well-vascularized cutaneous layer. The punctured place should be warmed, resulting in dilated blood vessels by consequently speeding up blood flood and lowering the difference between the arterial and venous gas.

In the present study, the impact of different types of face coverings on the acid–base balance and the function of respiratory track was analyzed. The results revealed the mask to be neutral while worn during normal everyday activities. There were no significant differences detected in both acid–base balance analysis and observed spirometry. However, there were significant differences in several noted parameters. Nevertheless, almost all obtained results were within the reference range. Since the fingertip capillary blood gas analysis is not the most precise method [17], we presume that aforementioned phenomena occurred due to the measurement bias. 

There is no doubt that face coverings create an additional obstacle for the air flow which needs to be overcome [18]. Therefore, it might be hypothesized that distal parts of the respiratory tract remain hyperventilated during the exposure time. The air flow could be hindered due to the accumulation of excretion which is constantly produced by the epithelium and not adequately removed due to the face covering. However, there were few differences between the first and second measurements in forced expiratory flow (FEF). There are several studies that investigate the impact of face masks on multiple parameters available. Lubrano and colleagues conducted a study in which the influence of surgical masks on respiratory function was assessed in the pediatric population. They intended to investigate if wearing surgical masks was associated with episodes of oxygen desaturation or respiratory distress. Obtained results revealed that the use of surgical masks among children was not associated with episodes of oxygen desaturation or the development of clinical signs of respiratory distress [19]. These results are coherent with those obtained in the present study, where a lack of desaturation, pulse fluctuation, and changes in blood pressure was observed. On the other hand, overweight or obese children were prone to develop respiratory distress exemplified by an increase in end-tidal carbon dioxide tension (PETCO2) as fast as wearing the mask and an increase in the PETCO2, pulse rate (PR), and respiration rate (RR) after the walking test. After the walking test, there was a significant correlation between PETCO2 and the body mass index [20].

In our study, the basic parameters assessing cardiac function were also determined. We did not notice statistically significant differences in the pulse and blood pressure. However, the measurements were taken during normal everyday activity. Egger et al. observed that the face mask can alter physical performance, especially at higher intensities in elite-level athletes [21]. It is important to note that the differences in physiological variables did not reach levels of clinical significance for cardiopulmonary dysfunction [22]. Results of another study revealed that surgical masks were associated with significant increased airway resistance, reduced oxygen uptake, and increased heart rate during continuous exercise [18]. In addition, Chew et al. demonstrated that the use of face masks can be associated with more self-reported cardiovascular symptoms such as dyspnea, fatigue, palpitations, and dizziness [23]. 

The results of the present study and abovementioned studies imply that face masks during normal activity do not impair respiratory and cardiovascular systems. Nevertheless, there are vulnerable populations, as well as special situations such as physical exercise or work, which require attention. In those situations, it should be considered that face coverings could not be neutral and may affect cardiac or respiratory functions. Moreover, symptoms of preexisting cardiac or respiratory conditions might be aggravated. 

In this paper, the microbial load on the particular types of protective masks commonly used worldwide during the COVID-19 pandemic was also evaluated. It is important to note that PPE external coverage with bacterial and fungal load may be strongly correlated with the quality of the air and the contamination of inanimate objects and surfaces within the environment in which mask users reside. The authors observed that high microbial load on the outer surface of surgical masks had a significantly positive correlation with the number of bacteria and fungi found in air pollution control samples of different wards. To reduce the microbial load of masks, the microbiological quality of the working environment should be improved.

This study also aimed to determine the role of face masks as potential vectors for the transmission of microorganisms, with a special emphasis on ESKAPE pathogens [8,24]. The moist and warm environment of its inner surface primarily provides the bacteria with suitable conditions for growth. In addition, the hospital environment itself, where the individuals using the face masks resided, increases the risk of microbial contamination of the outer side of such a face covering. Any interaction between the hands or the environment with the surface of face mask may result in its contamination [25]. Individuals touch their eyes up to 20 times per hour, also due to improperly fitted mask. This eventually results in the risk of hands becoming contaminated and the subsequent contamination to surroundings [26]. This definitely shows that face mask users should avoid circumstances when hands come into contact with the surface of the face mask and unconditionally respect the wearing time recommended by the manufacturer. The results obtained by Zhiquing et al. demonstrated that surgical mask surface contamination increases with a prolonged wearing time [14]. A somewhat different conclusion was reached by Nightingale et al. who observed that the duration of mask use did not correlate with bacterial contamination [27]. Nevertheless, from the research in this paper, it can be concluded that wearing the face mask for 3 h, as recommended by the manufacturer, is considered safe in terms of its microbial surface contamination. In the present study, we confirmed the lack of Gram-negative rods and Enterococcus presence on the examined face covering surfaces, unlike Nightingale et al. who determined the presence of these microorganisms on face coverings collected from a nursing home at the end of the user’s shift. The authors confirmed the presence of *S. aureus* in nearly 16% of the masks, which seems to be in accordance with the results obtained in this study. In the context of environmental contamination with *S. aureus* strains through the improper use and disposal of masks, our results indicate the potential risk of the transmission of strains within the hospital [27]. Health care staff are found to be involved in the cross-transmission of *S. aureus*; however, the routine screening of health care workers and the eradication of the *S. aureus* carrier state remain questionable [28]. 

Health care professionals play an important role in the cross-transmission of *S. aureus* either as a reservoir or as vectors. Patient colonization with *S. aureus* is associated with ward overcrowding, patient cohorting, and close contact between patients [29,30,31]. Shi et al. confirmed the *S. aureus* contamination of various items of hospital wards transmitted via the hands of hospital staff personnel [32]. Similar results were obtained by Allegranzi et al. and Gund et al. [33,34] who observed that the hands of personnel are an important transmitter of pathogens that also contaminate the healthcare environment. Nevertheless, the researchers emphasized that a strong awareness of workplace hygiene, routine control of the hospital settings, as well as sanitary regimens in wards at high risk of pathogen transmission are crucial in breaking the chain of pathogen dissemination through this route.

Although the study revealed no significant contamination of the mask’s surface with microorganisms other than *S. aureus* included in the ESKAPE group, it should be emphasized that reducing the touch while wearing the mask can significantly reduce the risk of cross-contamination.

Therefore, it can be concluded that the relatively low percentage of potentially dangerous pathogens (particularly from the ESKAPE group) on the outer surface of the mask may be a result of the high standards of hospital conditions in which the study was conducted. The maintenance of a high sanitary regime in the era of the pandemic not only concerns behavior in relation to limiting the transmission of SARS-CoV2, but is also related to an awareness of the spread of nosocomial and often multidrug-resistant pathogens that colonize patients and hospital staff. The results obtained in the present study indicated the presence of *S. aureus* on face mask surfaces in both nasal carriers and non-carriers of this microorganism; therefore, our study results raise awareness on the risk of face mask cross-contamination with *S. aureus* within the hospital environment.

No significant differences in the qualitative composition of microbial loads on face masks (surgical, FFP2, and FFP3 masks) were determined. Moreover, the culture technique used had no significant impact on the obtained results, where due to the specific structure, imprints were performed with FFP2 and FFP3 masks and surface stiffness, as well as the swabs from their both surfaces.

There are several statements on the potential correlation between the use of face masks and the contamination of yeasts and molds on their surfaces. Park et al. have suggested that the contamination of the mask’s surface with fungi increases with time of use [24]. All the study participants wore face masks for 4 h, which, according to available data, is the maximum time to decrease the bioburden of the mask surface [35]. Thus, the result of the present study contradicts this theory as none of these fungi were detected on any of the three types of masks used by the 50 individuals enrolled in this study. 

## 5. Conclusions

Face coverings worn during regular everyday activities appeared to not influence the acid–base balance and did not have an impact on the function of the respiratory tract. Nevertheless, in hospital settings, masks, regardless of their kind, should be worn and disposed with uttermost care, since they could be a potential vector of bacterial transmission, similarly to the hands of medical staff. The presence of *S. aureus* was confirmed in both nasal carriers and non-carriers, thus proving the cross-contamination and spread of this bacterium via hands. Furthermore, *S. aureus* contaminates the external and internal surfaces of face masks of each type, and is also transmitted via hands from external sources. Moreover, the absence of Gram-negative rods, which were considered the main hospital-acquired pathogens, demonstrate the good practices of infection control in examined respondents. The lack of yeasts was confirmed, which may serve as supporting information concerning the increased risk of superficial face mycoses, though most bacterial contamination originates from participant skin and respiratory tract microbiota. The 3 h exposure time is not sufficient for Gram-negative rods and mold contamination. 

## 6. Limitations

Our study provided interesting results. However, it has several drawbacks. In the first place, we did not perform arterial blood gas analysis because the Bioethical Committee approved capillary blood gas analysis due to its less invasiveness. This could influence results since capillary blood gas analysis is a less precise method. Moreover, not every participant, despite instructions, was able to achieve a forced expiratory time (FET) longer than six seconds, which diminished the mean FET in the study group. We performed spirometry, which is a reliable diagnostic tool; nevertheless, body plethysmography may provide more accurate results.

## Figures and Tables

**Figure 1 ijerph-20-02474-f001:**
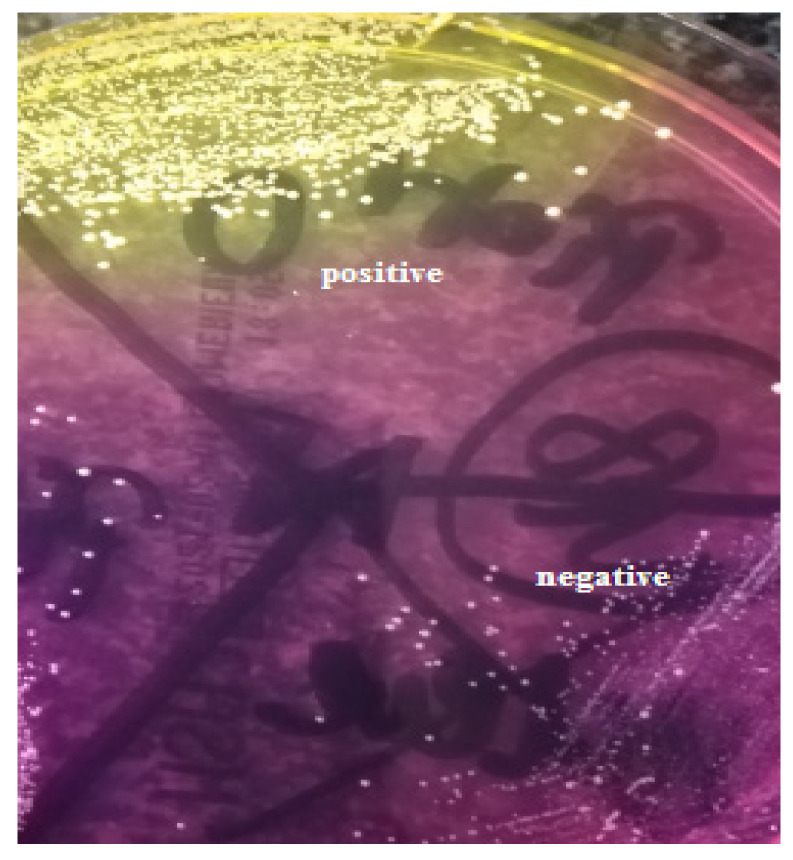
Mannitol salt agar culture result.

**Figure 2 ijerph-20-02474-f002:**
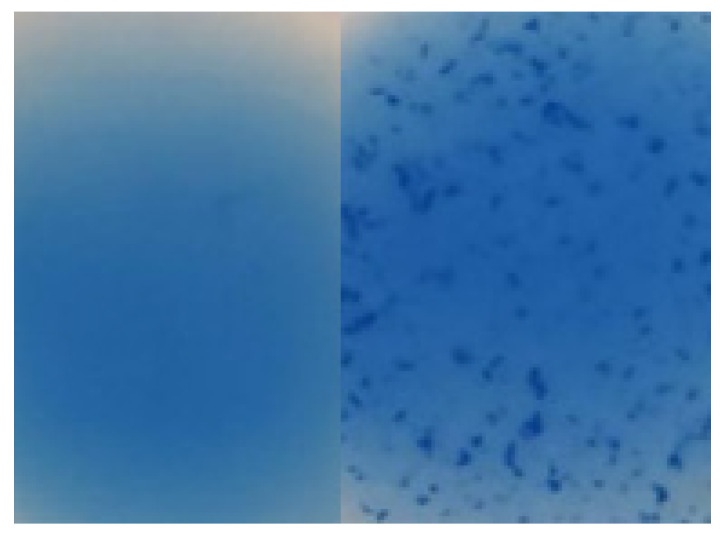
Staphaurex latex agglutination test results (left—negative, right—positive).

**Figure 3 ijerph-20-02474-f003:**
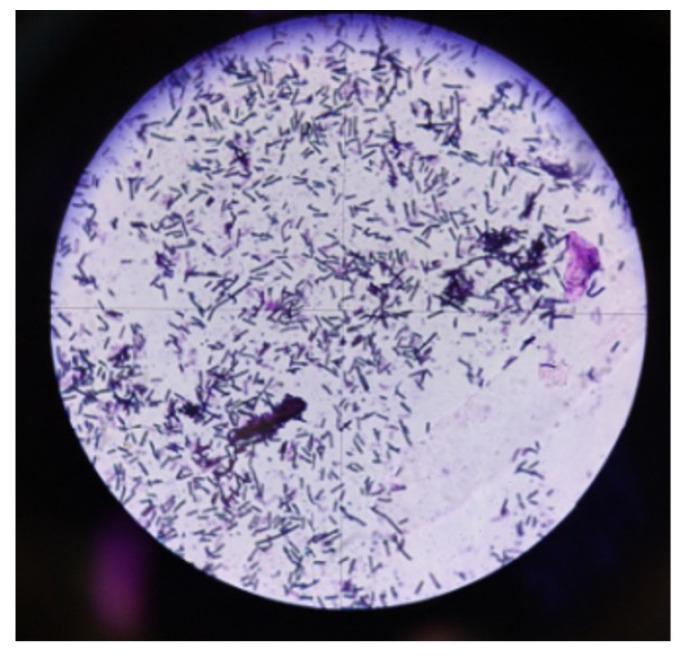
Gram-positive spore-forming rods.

**Table 1 ijerph-20-02474-t001:** Characteristics of the study group.

Parameter	N (%)/Mean ± SD
Sex (women/men)	38/12 (76/24)
Smoking (no/yes)	35/15 (70/30)
Chronic illnesses (no/yes)	27/23 (54/46)
Age	43.88 ± 14.52
BMI	26.49 ± 5.77
Weight	76.59 ± 19.97
Height	1.7 ± 0.1

**Table 2 ijerph-20-02474-t002:** Quantitative analysis of species detected on internal and external surface of a given face covering using the Wilcoxon signed-rank test. A *p* value < 0.05 was considered statistically significant.

**Surgical Mask Swab**	**External Swab**	**Internal Swab**	***p* Value**
	**Mean (Median)**	**Mean (Median)**	
CNS	1.5(1)	2.26(2)	0.000256
Aerobic spore- forming rods	0.07895(0)	0.05263(0)	0.7728
Gram (-) rods	Not detected	Not detected	-
*S. aureus*	0.1053(0)	0.2432(0)	0.1198
Oral Streptococcus	0.05263(0)	0.4474(0)	0.02074
Yeasts	Not detected	Not detected	-
Molds	Not detected	Not detected	-
**Surgical Mask Imprint**	**External Imprint**	**Internal Imprint**	***p* Value**
	**Mean (Median)**	**Mean (Median)**	
CNS	2.263(2)	2.595(3)	0.02402
Aerobic spore- forming rods	0.2105(0)	0.1081(0)	0.2755
Gram (-) rods	Not detected	Not detected	-
*S. aureus*	0.1842(0)	0.2162(0)	0.7656
Oral Streptococcus	0.1316(0)	0.2973(0)	0.04108
Yeasts	Not detected	Not detected	-
Molds	Not detected	Not detected	
**FFP2 Respirator**	**External Swab**	**Internal Swab**	***p* Value**
	**Mean (Median)**	**Mean (Median)**	
CNS	1.574(1)	2.021(2)	<0.001
Aerobic spore-forming rods	0.04255(0)	0.02128(0)	0.7728
Gram (-) rods	0.04255(0)	Not detected	-
*S. aureus*	0.2128(0)	0.1702(0)	0.6078
Oral Streptococcus	0.08511(0)	0.4468(0)	0.001422
Yeasts	Not detected	Not detected	-
Molds	Not detected	Not detected	-
**FFP3 Respirator**	**External Swab**	**Internal Swab**	***p* Value**
	**Mean (Median)**	**Mean (Median)**	
CNS	1.75(2)	2.073(2)	0.02819
Aerobic spore- forming rods	0.1	Not detected	-
Gram (-) rods	Not detected	Not detected	-
*S. aureus*	0.1(0)	0.2(0)	0.2795
Oral Streptococcus	0.025	0.4878	0.000544
Yeasts	Not detected	Not detected	-
Molds	Not detected	Not detected	-

**Table 3 ijerph-20-02474-t003:** *S. aureus* carrier state.

*S. aureus* Carrier State	
Yes	No
7	43

**Table 4 ijerph-20-02474-t004:** The presence of *S. aureus* on the surface of surgical, KN95, and FFP3 masks using Fisher’s exact test. *p* value < 0.05 was considered statistically significant.

**Surgical Mask**		
Carrier state	*S. aureus*	Frequency
NO	NO	25
YES	NO	2
NO	YES	7
YES	YES	3
*p* value	0.11	
**KN95**		
Carrier state	*S. aureus*	Frequency
NO	NO	33
YES	NO	3
NO	YES	7
YES	YES	4
*p* value	0.04	
**FFP3**		
Carrier state	*S. aureus*	Frequency
NO	NO	26
YES	NO	5
NO	YES	6
YES	YES	2
*p* value	0.62	

**Table 5 ijerph-20-02474-t005:** Results of capillary blood gas analysis and basic parameters for the surgical mask using the Wilcoxon signed-rank test. A *p* value < 0.05 was considered statistically significant. Statistically significant results are shown in bold.

	Before			After			*p* Value
	Mean	Median	SD	Mean	Median	SD	
Saturation	98.16	99	1.24	98.17	99	1.17	0.722
Pulse	81.45	78.5	15.57	80.14	80	11.69	0.5008
Systolic blood pressure	129.8	128.5	14.78	121.06	125	22.24	0.12
Diastolic blood pressure	82.79	82.5	9.68	79.03	78	6.53	0.12
pH	7.44	7.43	0.03	7.43	7.43	0.03	1
pCO2	37.85	38	2.69	37.44	37.5	3.56	0.4531
pO2	127.23	132	37.54	111.53	98	37.18	0.1191
Na+	137.12	137.5	2.53	137.38	137.5	2.85	0.5508
K+	5.54	5.4	0.98	5.55	5.25	0.86	0.5129
Ca2+	1.17	1.17	0.05	1.17	1.17	0.06	0.8754
Glu	94.85	92	17.62	91.34	89.5	12.70	0.2539
Lac	1.65	1.55	0.40	1.81	1.8	0.43	0.313
Hct	46.33	44.5	9.06	44.45	44	4.77	0.0171
Ca2+	1.19	1.18	0.05	1.19	1.18	0.07	0.9358
HCO3-	25.49	25.55	1.59	24.94	25.25	1.94	0.1959
HCO3std	26.02	26.05	1.25	25.56	25.75	1.30	0.1862
TCO2	26.65	26.65	1.65	26.09	26.45	2.03	0.1993
BEecf	1.28	1.55	1.81	0.66	1.1	2.01	0.2555
BE (B)	1.37	1.45	1.63	0.82	1	1.69	0.3006
SO2c	98.42	99	1.50	97.47	97.5	2.27	0.1432
Thbc	8.64	8.4	0.94	8.56	8.4	0.93	0.07918

**Table 6 ijerph-20-02474-t006:** Results of the blood gas examination and basic parameters of the FFP2 mask using the Wilcoxon signed-rank test. A *p* value < 0.05 was considered statistically significant. Statistically significant results are shown in bold.

	Before			After			*p* Value
Parameter	Mean	Median	SD	Mean	Median	SD	
Saturation	98.06	99	1.28	98.30	99	1.10	0.15
Pulse	80.70	78	14.38	79.81	78	13.90	0.61
Systolic blood pressure	132.36	128	15.01	132.93	132.5	15.25	0.28
Diastolic blood pressure	85.49	85	9.84	84.83	84	7.20	0.10
pH	7.43	7.43	0.03	7.43	7.43	0.02	0.934
pCO2	37.56	38	2.90	37.85	38	3.07	0.6379
pO2	121.54	115	40.80	109.79	94	40.64	0.207
Na+	137.41	137	2.55	137.21	137	2.40	0.2784
K+	5.97	5.7	1.23	5.45	5.3	0.80	0.008375
Ca2+	1.18	1.17	0.05	1.18	1.16	0.06	0.8151
Glu	101.10	94	20.08	94.36	93	18.90	0.08411
Lac	1.99	2	0.78	1.75	1.6	0.66	0.02211
Hct	44.38	44	4.52	43.89	44.5	4.37	0.01419
Ca2+	1.19	1.18	0.06	1.19	1.18	0.05	0.73
HCO3-	24.84	25.1	1.49	24.84	24.6	1.87	0.9288
HCO3std	25.51	25.7	1.09	25.44	25.3	1.28	0.6091
TCO2	26.03	26.4	1.56	26.01	25.8	1.94	0.943
BEecf	0.50	0.8	1.60	0.50	0.3	1.91	0.9404
BE (B)	0.66	0.9	1.39	0.63	0.5	1.64	0.7323
SO2c	97.90	99	2.10	97.03	98	2.68	0.2846
Thbc	9.49	8.4	5.90	10.37	8.55	11.80	0.04976

**Table 7 ijerph-20-02474-t007:** Results of the blood gas analysis and basic parameters of the FFP3 mask using the Wilcoxon signed-rank test. A *p* value < 0.05 was considered statistically significant. Statistically significant results are shown in bold.

	Before			After			*p* Value
Parameter	Mean	Median	SD	Mean	Median	SD	
Saturation	98.25	99	1.10	98.32	99	0.98	0.6925
Pulse	80.84	79	13.82	79.76	77.5	13.51	0.7885
Systolic blood pressure	130.2	128	13.37	128.9	127	12.68	0.6646
Diastolic blood pressure	82.23	82	7.41	80.38	79.5	8.14	0.343
pH	7.43	7.43	0.02	7.43	7.43	0.02	0.743
pCO2	37.13	37	3.01	37.08	36.5	3.37	0.8649
pO2	126.78	133	37.94	124	130.5	34.81	0.3868
Na+	136.75	137	2.71	137.34	137	2.69	0.227
K+	5.725	5.65	0.97	5.53	5.4	0.77	0.179
Ca2+	1.17	1.18	0.05	1.18	1.17	0.05	0.2482
Glu	96.5	95	12.71	98.97	90.5	30.52	0.675
Lac	1.87	1.75	0.59	1.84	1.7	0.71	0.9712
Hct	45.72	44	10.99	43.94	44	4.49	0.1945
Ca2+	1.19	1.18	0.05	1.19	1.18	0.05	0.06509
HCO3-	24.87	25.15	1.92	24.62	24.6	1.99	0.7102
HCO3std	25.56	25.75	1.36	25.33	25.5	1.33	0.4317
TCO2	26.00	26.3	1.99	25.76	25.8	2.08	0.7531
BEecf	0.61	0.9	2.05	0.31	0.3	2.03	0.5088
BE (B)	0.79	1.05	1.75	0.51	0.7	1.71	0.4028
SO2c	98.22	99	1.66	98.11	99	1.82	0.6093
Thbc	8.48	8.35	0.86	8.47	8.4	0.88	0.3297

**Table 8 ijerph-20-02474-t008:** Results of spirometry examination of the surgical mask using the Wilcoxon signed-rank test. A *p* value < 0.05 was considered statistically significant. Statistically significant results are shown in bold. Statistically significant results are shown in bold.

	Before			After			*p* Value
	Mean	Median	SD	Mean	Median	SD	
FVC	3.86	3.75	1.05	3.95	3.96	1.02	0.1883
FEV1	3.28	3.13	0.87	3.23	3.3	0.95	0.7218
FEV1/FVC (%)	83.99	83.6	7.62	81.19	82.9	9.32	0.5774
PEF	7.56	7.43	2.53	7.37	7.07	2.81	0.5128
FET	5.46	3.47	11.57	4.43	4.47	1.75	0.06768
FEF25	6.45	6.04	2.24	6.20	5.8	2.38	0.3982
FEF50	4.04	4.06	1.36	3.86	3.94	1.41	0.1745
FEF75	1.56	1.42	0.74	1.56	1.62	0.76	0.8327

**Table 9 ijerph-20-02474-t009:** Results of the spirometry examination of the FFP2 mask using the Wilcoxon signed-rank test. A *p* value < 0.05 was considered statistically significant. Statistically significant results are shown in bold.

	Before			After			*p* Value
Parameter	Mean	Median	SD	Mean	Median	SD	
FVC	3.99	3.74	1.06	3.99	3.82	1.03	0.7661
FEV1	3.33	3.21	0.97	3.34	3.15	0.95	0.7959
FEV1/FVC (%)	83.51	83.4	8.84	83.36	84.7	8.65	0.7311
PEF	7.19	6.76	2.54	7.50	6.77	2.52	0.02864
FET	4.13	4.29	1.92	4.02	3.7	1.99	0.6473
FEF25	6.48	5.95	2.31	6.41	6.05	2.17	0.899
FEF50	4.27	4.24	1.71	4.17	4.2	1.66	0.3913
FEF75	1.81	1.68	0.97	1.75	1.59	0.89	0.5625

**Table 10 ijerph-20-02474-t010:** Results of the spirometry examination of the FFP3 mask using the Wilcoxon signed-rank test. A *p* value < 0.05 was considered statistically significant. Statistically significant results are shown in bold.

	Before			After			*p* Value
Paramter	Mean	Median	SD	Mean	Median	SD	
FVC	3.94	3.85	1.01	3.89	3.78	1.00	0.9146
FEV1	3.31	3.25	0.93	3.25	3.18	0.92	0.2021
FEV1/FVC (%)	83.37	83.5	8.64	83.20	82.95	7.30	0.3657
PEF	7.58	6.95	2.73	8.88	6.81	9.96	0.9595
FET	3.99	3.56	2.19	4.14	3.86	1.89	0.4608
FEF25	6.48	6.07	2.51	6.26	5.99	2.44	0.4167
FEF50	4.063	3.965	1.636073	3.995897	4.19	1.52	0.5453
FEF75	1.64	1.52	0.92	1.60	1.4	0.87	0.3049

## Data Availability

This research received no external funding.

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
