# Peer review of "Medical Face Masks Do Not Affect Acid–Base Balance Yet Might Facilitate the Transmission of Staphylococcus aureus in Hospital Settings during the COVID-19 Pandemic"

_ijerph, 2023, doi:10.3390/ijerph20032474_

Round 1
Reviewer 1 Report
Piotr Ostrowski et al evaluated the rate of microorganism contamination of medical face masks of employees in the hospital and the impact of uninterrupted face cover with medical mask, on respiratory track function. And the authors expressed the opinion that medical face masks did not affect acid-base balance yet might facilitate the transmission of Staphylococcus aureus in hospital settings during COVID-19 pandemic.
1. The results do not support medical face masks might facilitate the transmission of Staphylococcus aureus in hospital.
2. A separate paragraph redescribes the results of the participants' infection.
3. There are some grammatical errors in the manuscript, such as” become commonplace phenomenon”(line 15)
Author Response
To Whom it may concern,
our response is attached in the file below. If you have any questions or concerns, please do not hesitate to contact us.
With best regards,
Piotr Ostrowski

Reviewer 2 Report
Row 19: The article was found inconsistent regarding the exposure time. Different parts of the article refer to the length of exposure time differently. 3 and 4 hours of exposure time were mentioned as well. Which is the correct information?
Row 95: 4 hours of exposure time is mentioned
Row 107: 3 hours of exposure time is mentioned
Row 149: Abbreviation CFU should be detailed: colony-forming unit
Row 165: The text refers to Fig. 4, but only two figures are included in the current article.
Row 178: Table 1. includes different results on the mean age (43.88) of participants. In this row: 48.88 years.
Row 187: The following table has no caption.
Row 214: The caption of Table 6 mentions FFP2 masks, like Table 5. The following paragraph, in Row 216 refers to FFP3 masks regarding Table 6.
Row 259: Abbreviation FEF should be detailed: forced expiratory flow
Row 270: Abbreviations PETCO2, PR, and RR should be detailed.
Row 273: This sentence is meaningless in current form: No alterations in ….
Row 311-312: This sentence, referring to the results of Zhiquing et al, should be signed with rank 14.
Row 315: 4 hours of exposure time is mentioned.
Row 329: The word ‘Li-sha’ can be deleted.
Row 332: Literature 34 has different authors than 33. They should be also mentioned in the text, not only Allegranzi and Pittel. Or they can be deleted and focusing on their findings would be enough.
Row 360: 4 hours of exposure time is mentioned
Row 377: The abbreviation RT should be detailed. 3 hours of exposure time is mentioned.
Row 385, 386: The abbreviation FET should be detailed.
Author Response
To Whom may it concern,
Our response is attached in the file below. If you have any questions or concerns, please do not hesitate to contact us.
With best regards,
Piotr Ostrowski
